# Minimally Invasive Hysterectomy Approaches: Comparative Learning Curves and Perioperative Outcomes of Robotic Versus V-NOTES Techniques

**DOI:** 10.3390/jcm14248743

**Published:** 2025-12-10

**Authors:** Sercan Kantarcı, Alaattin Karabulut, Uğurcan Dağlı, Batuhan Baykuş, Serhat Sarıkaya, Mehmet Özer, Alper İleri, Abdurrahman Hamdi İnan

**Affiliations:** 1Department of Obstetrics and Gynecology, University of Health Sciences, Tepecik Education and Research Hospital, 35020 Izmir, Turkey; 2Department of Obstetrics and Gynecology, University of Health Sciences, Erzurum City Hospital, 25240 Erzurum, Turkey

**Keywords:** robotic hysterectomy, V-NOTES, learning curve

## Abstract

**Objectives:** To compare perioperative outcomes and learning curves of robotic hysterectomy and transvaginal natural orifice transluminal endoscopic surgery (V-NOTES) hysterectomy performed for benign gynecological conditions in a high-volume tertiary center. **Methods:** This retrospective cohort study included 100 patients who underwent either robotic hysterectomy (n = 44) or V-NOTES hysterectomy (n = 56) between January 2024 and July 2025. Demographic data, perioperative parameters, and postoperative outcomes were collected. Learning curves were analyzed using cumulative sum (CUSUM) and quadratic regression models. **Results:** A total of 100 patients were included (44 robotic, 56 V-NOTES). Baseline demographics were comparable between groups. The postoperative hemoglobin decrease was significantly lower in the robotic group (0.96 ± 0.64 g/dL vs. 1.33 ± 0.93 g/dL, *p* < 0.05), whereas uterine weight was higher in the V-NOTES cohort (182.6 ± 125.9 vs. 123.2 ± 60.4 g, *p* < 0.05). Complication rates, including three bladder injuries in the V-NOTES group and one in the robotic group, showed no significant difference. Hospital stay was similar across groups. **Conclusions:** Both techniques are safe and effective. Robotic hysterectomy offers shorter operative time and less blood loss, while V-NOTES provides cosmetic and recovery advantages. Learning curve analysis indicates a longer adaptation period for V-NOTES, with anterior colpotomy as the most critical step, whereas robotic hysterectomy demonstrates a shorter and more straightforward learning process.

## 1. Introduction

Hysterectomy is one of the most commonly performed gynecological procedures worldwide, with approximately 20–40% of women undergoing it by the age of 60 [1,2]. Over recent decades, surgical approaches have evolved significantly from open abdominal techniques to minimally invasive methods aimed at reducing morbidity, enhancing recovery, and improving patient satisfaction. Vaginal hysterectomy (VH) has long been considered the gold standard for benign uterine conditions due to its shorter hospital stay, fewer wound complications, and better cost-effectiveness compared to abdominal hysterectomy. However, VH can be technically challenging in patients with non-prolapsed, enlarged uteri or a history of pelvic surgery [3].

The introduction of laparoscopic hysterectomy broadened the scope of minimally invasive surgery by offering improved visualization and expanded indications. It demonstrated advantages over open surgery in terms of reduced blood loss, less postoperative pain, and quicker recovery [4]. However, conventional laparoscopy is limited by its reliance on two-dimensional imaging, reduced instrument flexibility, and a steep learning curve [5]. In response to the technical constraints inherent in conventional laparoscopy such as limited instrument articulation, the absence of depth perception due to two-dimensional imaging, and the steep learning curve, robotic-assisted laparoscopic surgery emerged as a refined alternative following the FDA’s approval of the *da Vinci Surgical System*. This advanced platform integrates features like three-dimensional high-definition visualization, tremor filtration technology, and articulating robotic instruments that collectively enhance surgical precision and ergonomics. Nevertheless, despite offering comparable perioperative outcomes to standard laparoscopic hysterectomy in terms of safety and effectiveness, robotic-assisted procedures have been consistently associated with significantly prolonged operative times and substantially higher healthcare costs [6,7].

More recently, transvaginal natural orifice transluminal endoscopic surgery (V-NOTES) has emerged as an innovative option within minimally invasive gynecology. By utilizing the vaginal route as a natural access point, V-NOTES eliminates the need for abdominal incisions, offering enhanced cosmetic outcomes, reduced postoperative pain, and accelerated recovery [8]. In addition, V-NOTES has broadened the indications for vaginal surgery to include patients with larger uteri, obesity, prior cesarean deliveries, or nulliparity [9]. Despite these advantages, the procedure remains technically demanding due to limited working space and potential instrument collisions. Integration of robotic technology into V-NOTES has the potential to address these limitations by combining the advantages of both approaches [10]. Although current comparative data between robotic and V-NOTES hysterectomy remain limited, preliminary evidence suggests differences in operative time, blood loss, and complication rates. These insights highlight the need for further research to support evidence-based decision-making and improve surgical training in minimally invasive gynecology. Therefore, the present study aims to compare robotic and V-NOTES hysterectomy in terms of perioperative outcomes, complication rates, and learning curve characteristics in patients undergoing surgery for benign gynecological conditions.

## 2. Methods

This study was designed as a single-center retrospective observational cohort analysis. Medical records of patients who underwent elective robotic hysterectomy or V-NOTES hysterectomy for benign gynecological indications between January 2024 and July 2025 at a tertiary referral center were reviewed. A total of 112 patients were initially screened for eligibility. Inclusion criteria were age ≥ 30 years, having undergone robotic or V-NOTES hysterectomy, complete perioperative and postoperative data, and at least four weeks of follow-up. Exclusion criteria included pre- or postoperative histopathological evidence of malignancy, emergency surgery, history of prior pelvic radiation, body mass index (BMI) greater than 40, active pelvic or systemic illness that could interfere with postoperative outcomes, coagulopathy or current anticoagulant therapy, any cancer diagnosis, and severe cardiopulmonary comorbidities rendering the patient unsuitable for elective surgery. After applying these criteria, 100 patients were included in the final analysis and categorized into two groups: the robotic hysterectomy group (n = 44) and the V-NOTES hysterectomy group (n = 56). The study protocol was approved by the institutional ethics committee (Decision No: 2025/08-18) and all procedures adhered to the principles of the Declaration of Helsinki.

All procedures were performed by five surgeons with extensive expertise in minimally invasive gynecology, each with experience of more than 100 laparoscopic and more than 100 vaginal procedures. The primary determinant in the selection of surgical approach was the surgeon’s experience and technical preference. In addition, V-NOTES was not preferred in patients with a history of cervical surgery such as LEEP or in cases with a potential risk of malignancy, where the robotic approach was chosen instead. Operative time was defined as the interval between the first skin incision and the completion of skin closure. Intraoperative blood loss was calculated by comparing the preoperative hemoglobin value obtained within 15 days before surgery with the postoperative hemoglobin value measured six hours after the procedure.

Collected variables included demographic and clinical characteristics such as age, BMI, parity, previous surgery, and surgical indication; perioperative parameters including operative time, blood loss, uterine weight, and intraoperative complications; and postoperative outcomes including complications, hospital stay, and time to discharge. Complications were classified as major (bladder, bowel, or ureteral injury, or major hemorrhage) and minor (vaginal cuff hematoma, wound infection, or vesicovaginal fistula).

Statistical analyses were performed using SPSS version 26 (IBM Corp., Armonk, NY, USA). Continuous variables were summarized as mean ± standard deviation or median (minimum–maximum), and the normality of data distribution was assessed using the Kolmogorov–Smirnov test and visual inspection. Parametric variables were analyzed using Student’s *t*-test, while non-parametric variables were compared using the Mann–Whitney U test. Categorical variables were expressed as counts and percentages and compared using the chi-square or Fisher’s exact test, as appropriate. Learning curve evaluation was conducted through linear regression analysis of operative times according to case order, and CUSUM (cumulative sum) analysis was employed to determine breakpoints where surgical performance demonstrated significant improvement. A *p*-value < 0.05 was considered statistically significant.

### 2.1. V-NOTES Hysterectomy Procedure

All patients were placed in a low lithotomy position with approximately 20° Trendelenburg tilt to optimize visualization of the pelvic cavity. After aseptic preparation and bladder catheterization, a circumferential incision was made at the cervicovaginal junction using monopolar energy. The uterosacral ligaments were identified, carefully dissected, and divided under direct visualization to achieve adequate uterine mobility and exposure. This maneuver was critical for safe and effective placement of the transvaginal access platform, allowing proper alignment and stabilization.

Following insertion of the access platform, pneumoperitoneum was established with carbon dioxide insufflation at a pressure of 15 mmHg. A 10 mm 0° laparoscope was introduced for visualization, and the pelvic organs were systematically inspected. Sequentially, the broad, round, and utero-ovarian ligaments were coagulated and transected using bipolar energy and an advanced vessel sealing system, maintaining meticulous hemostasis throughout the procedure. When adnexectomy was indicated, the infundibulopelvic ligaments were similarly sealed and divided.

After complete detachment of the uterus, it was extracted transvaginally through the posterior colpotomy. In cases of limited vaginal space or larger uterine volume, morcellation under direct visualization was performed to facilitate removal while preserving safety. The pelvic cavity was then irrigated, inspected for hemostasis, and the vaginal cuff was closed with a single layer of delayed-absorbable continuous sutures. All instruments were withdrawn under direct vision, and the pneumoperitoneum was released before the procedure was completed.

This standardized technique aimed to ensure consistent exposure, optimal visualization, and reproducible perioperative outcomes across all surgeons involved.

### 2.2. Robotic Hysterectomy Procedure

All robotic hysterectomy procedures were performed using the da Vinci^®^ Surgical System (Si platform, Intuitive Surgical, Sunnyvale, CA, USA). Following the induction of general anesthesia, patients were positioned in the low lithotomy position with 25–30° Trendelenburg tilt to optimize bowel displacement and exposure of the pelvis. After aseptic preparation and bladder catheterization, pneumoperitoneum was established with carbon dioxide insufflation at a pressure of 15 mmHg through a Veress needle inserted at the umbilicus.

Four trocars were placed under direct laparoscopic visualization: one 12 mm umbilical camera port, two 5 mm robotic working ports placed bilaterally along the lower quadrants, and one 10 mm accessory port for the assistant. The robotic cart was then docked in a standardized pelvic orientation, and the surgeon proceeded from the console, utilizing three-dimensional high-definition visualization, which provided enhanced depth perception and precision during dissection.

The uterine vessels were carefully skeletonized, coagulated, and transected using bipolar energy and an advanced vessel sealing system, ensuring meticulous hemostasis. Subsequently, the cardinal, uterosacral, and remaining supporting ligaments were divided in a stepwise manner to achieve full uterine mobilization. When adnexectomy was indicated, the infundibulopelvic ligaments were also sealed and transected under direct vision.

In cases where the uterus was of normal size, the specimen was extracted transvaginally through a posterior colpotomy. For larger uteri, contained morcellation or endoscopic retrieval was performed within a protective bag to maintain minimal invasiveness and prevent tissue dissemination. The vaginal cuff was then closed intracorporeally using a continuous delayed-absorbable suture, ensuring accurate approximation, secure closure, and minimal tissue trauma.

Before completion, the pelvic cavity was irrigated and inspected to verify hemostasis. All trocars were removed under direct vision, pneumoperitoneum was released, and port sites were closed with absorbable sutures. This standardized robotic approach allowed consistent exposure, high procedural precision, and reproducible perioperative outcomes across all surgeons involved in the study.

## 3. Results

A total of 100 patients were included in the study, of whom 44 underwent robotic hysterectomy and 56 underwent V-NOTES hysterectomy. The baseline demographic and clinical characteristics of both groups are summarized in Table 1. No significant differences were observed in terms of age (52.16 ± 8.55 vs. 50.86 ± 6.75 years, *p* = 0.397), body mass index (27.23 ± 2.92 vs. 26.82 ± 3.61 kg/m^2^, *p* = 0.548), or parity (3.20 ± 2.11 vs. 2.91 ± 1.18, *p* = 0.380) between the robotic and V-NOTES groups, respectively. The distribution of comorbidities and previous abdominal surgeries was comparable, although the difference in prior cesarean section rates approached statistical significance (*p* = 0.054). Indications for surgery differed significantly between groups, with myoma uteri and treatment-resistant menometrorrhagia being more frequent in the V-NOTES cohort (*p* = 0.036).

The mean operative time was significantly shorter in the robotic group compared with the V-NOTES group (86.82 ± 21.33 vs. 95.0 ± 13.82 min, *p* < 0.05). The postoperative hemoglobin decrease was significantly lower in the robotic group (0.96 ± 0.64 g/dL vs. 1.33 ± 0.93 g/dL, *p* < 0.05). Mean uterine weight was significantly higher in the V-NOTES group (182.6 ± 125.9 g vs. 123.2 ± 60.4 g, *p* < 0.05). Minor complications were observed in 2.3% of robotic cases and 7.1% of V-NOTES cases, with no statistically significant difference (*p* = 0.179). Major complications occurred in 2.3% and 5.4% of robotic and V-NOTES procedures, respectively, without significant difference (*p* = 0.435). The most frequent minor complication was urinary tract infection (n = 4) in the V-NOTES cohort, whereas cuff hematoma was observed in both groups. Major complications were rare and occurred in 4 patients (4.0%) in total, with three bladder injuries in the V-NOTES group and one in the robotic group; the difference was not statistically significant (*p* = 0.435). No ureteral or bowel injuries were identified in either cohort. The mean hospital stay was similar between groups (1.84 ± 1.33 vs. 1.66 ± 1.15 days, *p* = 0.507) (Table 2).

V-NOTES hysterectomy demonstrated a four-phase learning curve with breakpoints at the 14th, 25th, and 40th cases, with anterior colpotomy as the most critical step (Figure 1)**,** whereas robotic hysterectomy exhibited a two-phase curve with a breakpoint at the 10th case, indicating a shorter and more predictable learning process (Figure 2).

## 4. Discussion

Minimally invasive gynecologic surgery has rapidly expanded worldwide over the past decades, driven by the need to reduce surgical morbidity, accelerate recovery, and improve patient satisfaction [11]. While laparoscopic hysterectomy represented a major milestone in this evolution, its inherent limitations such as two-dimensional imaging and restricted instrument articulation created the need for further innovation. Robotic-assisted surgery, introduced into gynecology after FDA approval of the da Vinci Surgical System in 2005, was developed to overcome these technical challenges by providing three-dimensional visualization, tremor filtration, and enhanced instrument dexterity [12]. More recently, transvaginal natural orifice transluminal endoscopic surgery (V-NOTES) has emerged as another innovative approach, utilizing the vaginal route as a natural access point and eliminating abdominal incisions, thereby offering superior cosmetic outcomes, reduced postoperative pain, and shorter recovery times [13]. Against this background, the present study was designed to compare the perioperative outcomes, complication profiles, and learning curves of robotic and V-NOTES hysterectomy, with the hypothesis that each technique carries unique advantages and limitations that may guide optimal surgical selection.

Baseline characteristics such as age, BMI, and parity were comparable between the groups, supporting the validity of perioperative comparisons. However, surgical indications differed significantly, with myoma uteri and treatment-resistant menometrorrhagia being more frequent in the transvaginal endoscopic group, whereas endometrial hyperplasia and persistent cervical dysplasia were more common in the robotic abdominal cohort. This distribution suggests that patient selection criteria may have influenced the choice of surgical approach and should be considered when interpreting perioperative outcomes. In particular, although cervical dysplasia is not regarded as an absolute contraindication to natural orifice access, patients with a history of cervical procedures such as LEEP or conization may present technical challenges during anterior and posterior colpotomies, which could explain the lower utilization in this subgroup. Malignant cases were not included in our study; however, in patients with atypical hyperplasia or dysplastic lesions carrying a potential risk of malignancy, we preferred the robotic abdominal approach, given its ability to provide a wider surgical field and facilitate comprehensive oncologic assessment. Nevertheless, previous reports have demonstrated that hysterectomy for endometrial cancer can also be successfully performed using the V-NOTES technique [14].

Although comparative studies between Total Laparoscopic Hysterectomy (TLH) and V-NOTES are more prevalent in the literature, direct comparisons with robotic hysterectomy remain scarce. In our study, the transvaginal approach was associated with a shorter mean hospital stay compared with the robotic technique (1.66 vs. 1.84 days; Table 2), although this difference did not reach statistical significance. This finding aligns with previous reports indicating reduced hospitalization after V-NOTES compared with TLH [15,16]. Taken together, these results suggest that the natural orifice route may facilitate faster recovery and earlier discharge; however, further investigations are warranted to determine whether these advantages are consistently maintained when directly compared with robotic hysterectomy.

In our study, the mean operative time was significantly shorter in the robotic abdominal hysterectomy group compared with V-NOTES (86.82 vs. 95.0 min, *p* < 0.05). This finding contrasts with many previous reports suggesting that robotic surgery may require longer operative times due to docking and setup steps. In our cohort, however, the robotic approach demonstrated a shorter operative time, likely reflecting institutional workflow efficiency and team proficiency. The higher mean uterine weight in the V-NOTES group may have contributed to increased blood loss, greater technical complexity, and longer operative time, particularly during the anterior colpotomy and specimen extraction steps. This effect is more pronounced in the early learning phase, as larger uteri are technically more challenging to extract transvaginally.

Conversely, postoperative decrease in hemoglobin level was significantly lower in the robotic group. (0.96 vs. 1.33 g/dL; *p* < 0.05). Similar findings have been reported in the literature, where robotic hysterectomy, compared with conventional TLH, has been shown in many studies to be associated with reduced intraoperative blood loss [17,18]. This advantage is attributed to three-dimensional visualization, tremor filtration, and articulated instruments that enable more precise dissection and vascular control. In our cohort, a similar trend was observed, with blood loss significantly lower in the robotic abdominal hysterectomy group compared with the V-NOTES group. Overall, our results suggest that despite the longer operative time, robotic hysterectomy may reduce blood loss not only compared with conventional laparoscopy but also with natural orifice access, thereby supporting the need to individualize the surgical approach according to patient characteristics and surgical complexity. Advances in visualization technologies, including 3D and high-definition imaging, have been shown to shorten the learning curve and enhance precision. A recent systematic review demonstrated that three-dimensional laparoscopy leads to better task accuracy and fewer errors among trainees compared to conventional two-dimensional systems [19]. Recent evidence suggests that collective team experience, rather than individual proficiency alone, plays a crucial role in improving surgical efficiency and institutional cost-effectiveness during the learning process of robot-assisted hysterectomy [20].

The only major complication identified in this cohort was bladder injury, occurring in one patient (2.3%) in the robotic hysterectomy group and in three patients (5.4%) in the V-NOTES group, with no statistically significant difference between the approaches. These findings are consistent with the literature, which shows that neither technique increases the risk of major morbidity. Swenson et al. reported an overall complication rate significantly lower with robotic hysterectomy compared with other minimally invasive approaches (3.5% vs. 5.6%; *p* = 0.01) [21]. Likewise, in a large series of 4565 procedures performed via V-NOTES, the perioperative complication rate was 3.6%, including 60 bladder injuries and one ureteral injury among the urological complications [22]. Complication rates in our series were comparable to those previously reported; however, although not statistically significant, the transvaginal approach demonstrated a higher overall rate, which may be explained by the fact that anterior colpotomy was considered the most critical and time-limiting step until sufficient experience was achieved. Moreover, in patients with a history of cesarean section, port placement and anterior colpotomy represented the most technically demanding and decisive steps of the procedure.

Our CUSUM analysis demonstrated distinct differences in the learning process of the two minimally invasive surgical approaches. CUSUM breakpoints were defined at the points where the slope of the cumulative deviation curve changed markedly, indicating improved efficiency. These points were further confirmed statistically using quadratic regression analysis to ensure consistency between visual and analytical interpretation. In V-NOTES hysterectomy, the learning curve consisted of four phases: Phase I (cases 1–14) represented the adaptation period with prolonged operative times; Phase II (cases 15–25) showed improvement with shorter times due to increasing technical proficiency; Phase III (cases 26–40) indicated a plateau phase with stabilization and reduced variability; and Phase IV (cases 41–55) reflected advanced proficiency with a renewed decrease in operative times. These findings highlight the importance of structured training and appropriate case selection during the early adoption of minimally invasive hysterectomy techniques. For V-NOTES, careful selection of cases with smaller uterine size and favorable vaginal access may facilitate a smoother learning process. In contrast, the shorter and more predictable learning curve of robotic hysterectomy may provide a safer and more standardized platform for surgeons in their initial experience.

Similarly, Charles et al. reported a four-phase learning process in a single-surgeon experience, demonstrating that basic proficiency was achieved after approximately 12 cases, while mastery was attained after 53 cases [23]. Huang et al. also described a four-phase process in V-NOTES, showing that adaptation was completed after the first 8 cases following the introduction of a standardized operating procedure (SOP), with further efficiency gains observed after 20 cases [24].

A key strength of this study is that it provides one of the largest early comparative series in the literature, including the first 100 consecutive cases from a high-volume tertiary center. All surgeons participating in this study were experienced in both conventional laparoscopic and vaginal hysterectomy, ensuring comparable baseline proficiency. However, the robotic and V-NOTES procedures analyzed here represent the early experience of our team with these specific techniques, during the initial institutional adoption phase. Therefore, the shorter operative times observed in the robotic group likely reflect system setup and coordination factors rather than differences in surgical skill. The main limitation of this study is its retrospective design, which may affect the uniformity of case selection and limit control over intraoperative decision-making. Furthermore, the single-center and single-team setting restricts generalizability, and the absence of long-term outcomes such as quality of life or late complications should be addressed in future prospective studies. As our institution does not have a single-port robotic platform, we compared the two most commonly performed minimally invasive techniques for similar benign indications multiport robotic hysterectomy and single-port V-NOTES hysterectomy. This series represents early-phase laparoscopic and V-NOTES hysterectomy cases, consistent with the bladder injury rates reported in the previous literature. The slightly higher complication rate observed in the V-NOTES group may be related to the anterior colpotomy step during the initial learning phase. Additionally, since five experienced surgeons performed the procedures and the choice of technique was mainly based on each surgeon’s expertise and preference, variations in surgical selection and learning progression among surgeons might have influenced operative time and complication patterns.

In the V-NOTES group, the CUSUM analysis was conducted based on operative time rather than complication rates. The curve demonstrated that the initial 14 cases represented the adaptation phase, after which operative times stabilized, indicating improved procedural efficiency. These findings suggest that approximately 15–20 cases are required to achieve a safe and proficient workflow, reflecting the collective experience of five surgeons working within the same minimally invasive gynecology unit. While numerous studies have compared V-NOTES with conventional total laparoscopic hysterectomy (TLH), direct comparisons with robotic hysterectomy remain scarce. Our study addresses this gap by analyzing the learning dynamics of both techniques. In the natural orifice approach, anterior colpotomy emerged as the most critical and time-limiting step, where longer operative times tended to cluster during the early adaptation period. Importantly, once the CUSUM-defined breakpoints were surpassed, both V-NOTES and robotic hysterectomy were performed safely and efficiently, demonstrating that a structured, multi-surgeon learning process can standardize performance and optimize surgical outcomes at the institutional level. While both techniques demonstrated comparable safety and efficacy, cost considerations remain an important determinant in clinical and institutional decision-making. Robotic surgery is associated with higher equipment and maintenance costs, whereas V-NOTES offers advantages in terms of reduced resource utilization and potentially lower overall costs.

## Figures and Tables

**Figure 1 jcm-14-08743-f001:**
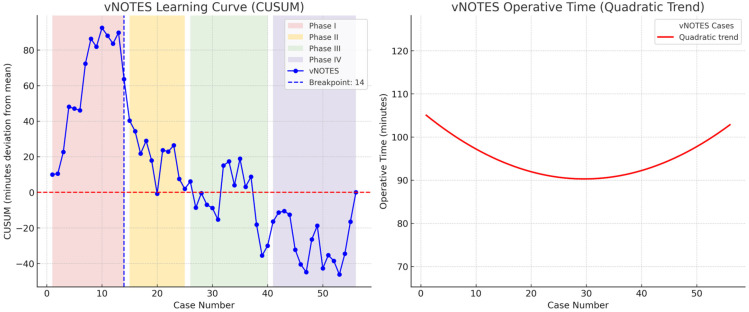
Cumulative sum (CUSUM) learning curve and quadratic regression trend for V-NOTES procedures. The CUSUM learning curve (**left panel**) demonstrates a four-phase process in the acquisition of surgical proficiency with V-NOTES hysterectomy. During the initial adaptation period (cases 1–14), the curve shows a steep upward trajectory, reflecting prolonged operative times. This is followed by a downward slope between cases 15 and 25, indicating progressive improvement and increasing technical mastery. Between cases 26 and 40, the curve enters a plateau phase in which operative times become more stable and variability decreases. Finally, after case 41, a renewed downward trend is observed, consistent with advanced proficiency and improved surgical efficiency. The quadratic regression trend (**right panel**) confirms this pattern by illustrating an initial decline in operative times, followed by a subsequent rise likely attributable to increased case complexity or inter-patient variability. These findings emphasize that anterior colpotomy represents the most critical and time-limiting step during the early phase of V-NOTES, but once the CUSUM-defined breakpoints are surpassed, the procedure can be performed safely, reliably, and with consistent efficiency.

**Figure 2 jcm-14-08743-f002:**
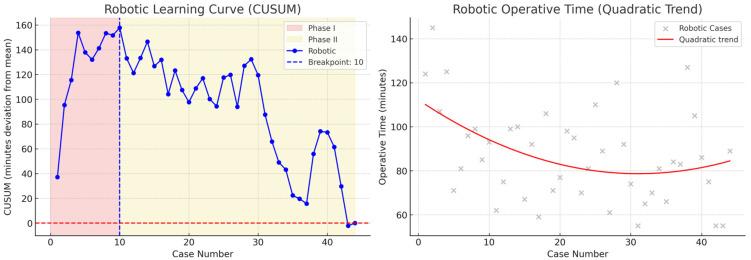
CUSUM learning curve and quadratic regression trend for robotic procedures. CUSUM learning curve and quadratic regression trend for robotic procedures. The CUSUM analysis identified two distinct phases. Phase I (cases 1–10) showed a progressive upward trend, indicating longer operative times during the initial learning period. Phase II (cases 11 onwards) revealed a consistent downward trajectory, suggesting achievement of procedural proficiency and improvement in operative performance. Unlike the V-NOTES group, no extended plateau or advanced phase was clearly observed, which is consistent with a shorter learning curve for robotic surgery. The quadratic regression demonstrated a gradual decrease in operative times across the series, with minor fluctuations reflecting case variability.

**Table 1 jcm-14-08743-t001:** Baseline demographic and clinical characteristics of patients undergoing robotic and V-NOTES hysterectomy.

Characteristic	Robotic (n = 44)	V-NOTES (n = 56)	*p*-Value
Age (years), mean ± SD	52.16 ± 8.55	50.86 ± 6.75	0.397
BMI (kg/m^2^), mean ± SD	27.23 ± 2.92	26.82 ± 3.61	0.548
Parity, mean ± SD	3.20 ± 2.11	2.91 ± 1.18	0.380
Comorbidities, n (%)			0.206
–None	20 (45.5)	15 (26.8)	
–Diabetes mellitus	5 (11.4)	8 (14.3)	
–Hypertension	7 (15.9)	19 (33.9)	
–Thyroid disease	5 (11.4)	5 (8.9)	
–Other	6 (13.6)	9 (16.1)	
Previous abdominal surgery, n (%)			0.054
–None	23 (52.3)	27 (48.2)	
–Appendectomy	1 (2.3)	7 (12.5)	
–Cesarean section	12 (27.3)	7 (12.5)	
–Cholecystectomy	6 (13.6)	6 (10.7)	
–Other	2 (4.5)	9 (16.1)	
Indication for surgery, n (%)			<0.05
–Myoma uteri	5 (11.4)	16 (28.6)	
–Ovarian cyst	0	2 (3.6)	
–Prolapse	12 (27.3)	10 (17.9)	
–Endometrial hyperplasia	8 (18.2)	4 (7.1)	
–Chronic pelvic pain	2 (4.5)	4 (7.1)	
–Treatment-resistant menometrorrhagia	11 (25.0)	18 (32.1)	
–Persistent cervical dysplasia	6 (13.6)	1 (1.8)	
–Endometriosis	0	1 (1.8)	

**Table 2 jcm-14-08743-t002:** Perioperative outcomes of robotic and V-NOTES hysterectomy.

Outcome	Robotic (n = 44)	V-NOTES (n = 56)	*p*-Value
Operative time (minutes), mean ± SD	86.82 ± 21.33	95.0 ± 13.82	<0.05
Postoperative hemoglobin decrease (ΔHb, g/dL), mean ± SD	0.96 ± 0.64	1.33 ± 0.93	<0.05
Uterine weight (g), mean ± SD	123.2 ± 60.4	182.6 ± 125.9	<0.05
Minor complications, n (%)			0.179
–None	43 (97.7)	52 (92.8)	
–Cuff hematoma	1 (2.3)	3 (5.3)	
–Fistula	0	1 (1.7)	
Major complications, n (%)			0.435
–None	43 (97.7)	53 (94.6)	
–Bladder injury	1 (2.3)	3 (5.4)	
Hospital stay (days), mean ± SD	1.84 ± 1.33	1.66 ± 1.15	0.507

## Data Availability

The data that support the findings of this study are available from the corresponding author upon reasonable request.

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
