# Peer review of "Minimally Invasive Hysterectomy Approaches: Comparative Learning Curves and Perioperative Outcomes of Robotic Versus V-NOTES Techniques"

_jcm, 2025, doi:10.3390/jcm14248743_

Round 1

Reviewer 1 Report

Comments and Suggestions for Authors

Detailed Peer Review Comments for the Manuscript Comparing Robotic Laparoscopic and V-NOTES Hysterectomy

Overall Recommendation: Major Revision Required
The primary concerns lie in the clarity of the study design, consistency of the data, and depth of the analysis.

Overall Impression
This study aims to compare the perioperative outcomes and learning curves of robotic laparoscopic hysterectomy and V-NOTES hysterectomy for benign gynecological conditions. The research topic is clinically relevant, as direct comparative data between these two advanced minimally invasive techniques remain limited. The use of CUSUM analysis for the learning curve is a notable strength.

However, the manuscript currently presents serious issues regarding internal inconsistencies (especially concerning key outcome data), ambiguous methodological descriptions, and insufficient analytical depth. These problems significantly compromise the reliability and interpretability of the conclusions. 

Major Issues Requiring Revision

1. Significant Concerns Regarding Study Design and Data Consistency

· Issue 1: Mismatch in Study Nature and Timeframe
  · Problem: The design involves a multi-dimensional comparison: multiport robotic abdominal surgery versus single-port VNOTES surgery, effectively comparing multiport vs. single-port and robotic vs. conventional laparoscopy. With multiple surgical teams involved, operative times and outcomes are subject to variation. The occurrence of 4 bladder injuries in 100 cases raises the question of whether the complication rate is higher than expected for these procedures.
  · Suggestion: All date-related information must be corrected to align with the actual study period. Citation formats need updating to the correct year and volume. The rationale behind the complex comparative design and the high complication rate should be explicitly addressed and justified.
· Issue 2: Contradictory Key Outcome Data
  · Problem: This is the most severe issue. The abstract claims "shorter operative time" for the robotic group, but raw data in the results (lines 133-134) show "86.82 ± 21.33 min for robotic vs. 95.0 ± 13.82 min for V-NOTES," with the corresponding p-value reported as p=0.507 (non-significant) in the text, yet as p<0.05 (significant) in Table 2 and elsewhere in the text. Similarly, inconsistencies exist for the p-value of blood loss between the abstract (p=0.047) and Table 2 (p<0.05).
  · Suggestion: All data throughout the manuscript, including the main text, tables, and abstract, must be thoroughly checked and unified. Confirm the correct p-values and ensure descriptions of core outcomes like operative time and blood loss are perfectly consistent with the statistical results. The current contradictions prevent readers from drawing reliable conclusions.
· Issue 3: Questionable Unit for Blood Loss
  · Problem: Blood loss is reported in "g/L," which is unconventional. Blood loss is typically reported in milliliters (mL), or as hemoglobin change (ΔHb) in g/dL.
  · Suggestion: Clarify and standardize the unit. If reporting hemoglobin change, using g/dL is advised. If reporting estimated blood volume, use mL. Ensure unit consistency throughout the manuscript.

2. Methodological Section Requires Clearer Description

· Issue 4: Unclear Study Group Designations
  · Problem: The abstract uses "Group I" and "Group II," while the methods section uses "Group A" and "Group B." This inconsistency causes confusion.
  · Suggestion: Use "Robotic Group" and "V-NOTES Group" consistently throughout the manuscript, avoiding Roman numerals or letters for group names.
· Issue 5: Lack of Details on Sample Size Calculation and Learning Curve Analysis
  · Problem: The manuscript does not mention the basis for sample size calculation. The CUSUM analysis for the learning curve lacks key parameters (e.g., target operative time, decision interval).
  · Suggestion: Explain how the sample size was determined (e.g., based on power analysis or all available cases during the period). Supplement the specific parameters used in the CUSUM analysis to demonstrate the objectivity of breakpoint identification.
· Issue 6: Confounding from Patient Allocation and Surgeon Experience
  · Problem: The methods state that surgeries were performed by five experienced surgeons but do not specify the number of cases per surgeon for each technique or their individual experience levels. The primary basis for technique selection being "surgeon's experience and preference" introduces significant selection bias, which is corroborated by the baseline differences in uterine weight and surgical indications between the groups.
  · Suggestion: The impact of this resulting selection bias on the interpretation of results must be thoroughly discussed in the limitations. Ideally, provide the case distribution per surgeon.

  1. Results and Discussion Sections Require Deepening

  • Issue 7: Inadequate Discussion of the Impact of Baseline Imbalance
  • Problem: The V-NOTES group had significantly larger uterine weights and different surgical indications. This likely influenced primary outcomes such as operative time and blood loss.
  • Suggestion: The discussion must acknowledge this limitation and explore how the difference in uterine weight might have confounded the comparison between techniques (e.g., a heavier uterus might make V-NOTES more challenging, potentially prolonging time).
  • Issue 8: Insufficient Clinical Interpretation of Learning Curve Analysis
  • Problem: Only the breakpoints of the learning curve are reported, without in-depth analysis of their clinical significance. For instance, were complications (especially bladder injuries) more concentrated during the initial V-NOTES learning phase (first 14 cases)?
  • Suggestion: Correlate the learning curve with complication rates. Discuss the implications of these breakpoints for training planning—how many cases are required to achieve a safe and efficient level?
  • Issue 9: Logical Flaws in the Discussion Argument
  • Problem: The discussion attempts to attribute longer robotic operative times to the "learning curve," yet the methods section clearly states all surgeons were already "experienced." This is a clear contradiction.
  • Suggestion: Re-examine and revise the arguments in the discussion. If the robotic procedures in this study were indeed on the learning curve, this needs explicit clarification. If the surgeons were already proficient, other plausible explanations (e.g., case complexity) should be sought, or the speculation should be removed.

Overall Recommendation

Major Revision.

The core concept of the study is novel, but its scientific value is currently overshadowed by severe data inconsistencies and methodological flaws.

The authors must:

  1. Thoroughly resolve all data contradictions, particularly the values and p-values for operative time and blood loss.
  2. Clarify and correct the study's timeframe and copyright information.
  3. Strengthen the methodology section, specifically regarding patient allocation, learning curve analysis, and statistical methods.
  4. Deepen the discussion, fully considering the impact of baseline imbalances on the results and providing a more profound interpretation of the learning curve's clinical significance.

After careful revision and adequate response to all the points raised above, this study has the potential to make a valuable contribution to the field.

Reviewer 2 Report

Comments and Suggestions for Authors

The manuscript addresses an important and timely topic in minimally invasive gynecologic surgery, comparing robotic and V-NOTES hysterectomy in terms of perioperative outcomes and learning curve characteristics. The paper is generally well written, and the results contribute valuable data to an area where comparative evidence remains limited. However, several aspects require clarification or further elaboration to improve the scientific rigor and reproducibility of the study.

Comments

  • The study is described as retrospective, but additional information is needed on patient selection and allocation to the two surgical groups. Please specify whether patient preference, anatomical factors, or surgeon availability influenced the choice of surgical approach.
  • Describe the two types of surgery in more detail.

  • It would be helpful to include a flowchart (CONSORT-style) showing patient inclusion and exclusion to ensure transparency.

  • The use of both CUSUM and quadratic regression models is appropriate. However, please explain the rationale for choosing the breakpoint cutoffs (14th, 25th, and 40th cases for V-NOTES; 10th for robotic). Were these defined statistically or visually?

  • Include the number of surgeons contributing to each learning curve and discuss whether inter-operator variability could have influenced the results.

  • The authors report that the robotic group had lower estimated blood loss, yet uterine weight was significantly higher in the V-NOTES cohort. This may confound the comparison and should be discussed further.
  • The description of statistical tests is adequate but could be expanded to specify which variables were treated as parametric vs. non-parametric.
  • Consider including effect sizes (e.g., Cohen’s d or odds ratios) to complement p-values and better convey clinical relevance.
  • The p-value for the difference in operative time and hospital stay is identical (p = 0.507); however, the former is reported as statistically significant, while the latter is not.
  • The discussion could benefit from a more detailed reflection on how these findings might influence surgical training and case selection for new surgeons adopting V-NOTES or robotic approaches.
  • The authors may wish to briefly address cost-effectiveness, as this factor often guides institutional decision-making between robotic and transvaginal endoscopic platforms.
  • Please proofread the manuscript for minor grammatical issues and inconsistencies in terminology (e.g., “robotic abdominal hysterectomy” vs. “robotic hysterectomy”).
  • Some data points (e.g., p-values for operative time) appear inconsistent between the text and tables—verify accuracy.

  • Figures 1 and 2 would benefit from improved labeling and clearer legends explaining the meaning of each phase in the learning curve.

  • In the “Methods” section, specify the energy sources used (bipolar, monopolar, vessel sealing system) in both techniques for better comparability.

  • Reference formatting should be standardized per journal style, particularly regarding spacing and punctuation. Add these references in the discussion:  PMID: 36811183, PMID: 41042398
